# Recruitment Patterns and Environmental Sensitivity of Glass Eels of *Anguilla japonica* in the Yangtze Estuary, China

**DOI:** 10.3390/biology13010056

**Published:** 2024-01-20

**Authors:** Hongyi Guo, Xuguang Zhang, Ya Zhang, Wenqiao Tang, Kai Liu

**Affiliations:** 1College of Fisheries and Life Science, Shanghai Ocean University, Shanghai 201306, China; hy-guo@shou.edu.cn (H.G.); zhangya@shou.edu.cn (Y.Z.); 2Engineering Technology Research Center of Marine Ranching, Shanghai Ocean University, Shanghai 201306, China; zhang_xuguang@163.com; 3Shanghai Universities Key Laboratory of Marine Animal Taxonomy and Evolution, Shanghai Ocean University, Shanghai 201306, China; 4Key Laboratory of Freshwater Fisheries and Germplasm Resources Utilization, Ministry of Agriculture and Rural Affairs, Freshwater Fisheries Research Center, Chinese Academy of Fishery Sciences, Wuxi 214081, China

**Keywords:** *Anguilla japonica*, ARIMA model, generalized additive models, glass eel recruitment, seasonal-trend decomposition, Yangtze River estuary

## Abstract

**Simple Summary:**

The Japanese eel, a high-valued commercial fish species in the Yangtze River ecosystem, faces a significant decline in the resource. In the present study, we reported our 11-year investigation to track the migratory patterns of glass eels of *Anguilla japonica* into the Yangtze estuary. We identified two distinct annual peak arrival times which coincide with cooler water temperatures, i.e., the first peak at 6–8 °C and the second at 10–12 °C, both in sync with the strong spring tides of the lunar cycle. Despite rebounded population in a few years, an auto-regressive integrated moving average model analysis reveals a concerning trend: since 2016, rising temperatures have been associated with delayed recruitment. This change poses a threat to their migratory routes and life cycle stability. Underlining the critical nature of our research, our findings call for immediate conservation efforts to counteract the impact of climate change on the eel and to safeguard the biodiversity of the river ecosystem, a valuable asset to society.

**Abstract:**

The decline of Japanese eel (*Anguilla japonica*) populations in the Yangtze River estuary represents a critical conservation concern. Eleven-years of daily catch data during recruitment periods (i.e., January–April, 2012–2022) indicate that annual catch averaged from 153 to 1108 eels, and show a bimodal pattern in glass eel arrivals. Utilizing seasonal-trend decomposition and generalized additive models, we demonstrated a strong correlation between catch abundance, optimal water temperatures, and lunar cycles. An auto-regressive integrated moving average (ARIMA) model predicts an increase in glass eel numbers for 2023–2024 but also points to a concerning trend of delayed recruitment timing since 2016, attributable to the 0.48 °C per decade rise in sea surface temperatures. This delay correlates with a significant decrease in the average body weight of glass eels, suggesting potential energy deficits that may hinder successful upstream migration. This study not only furthers our understanding of glass eel recruitment dynamics but also underscores the urgent need for targeted conservation measures. Additionally, it highlights the importance of sustained, detailed monitoring to mitigate the detrimental effects of climate change on these eels, vital for preserving the Yangtze River’s ecological integrity.

## 1. Introduction

The sustainability of global fishery resources is jeopardized by increasing environmental stressors and habitat degradation, primarily due to diverse human activities [1,2,3]. Migratory fish species, including those of the genus *Anguilla*, are especially vulnerable to heightened human threats, such as habitat loss and the disruption of migration pathways, which significantly affect their migration behaviors and population dynamics [4,5,6]. In addition, environmental drivers such as climate change, habitat alteration, and overfishing collectively exacerbate the challenges faced by the Japanese eel (*Anguilla japonica*) [7]. These factors have a profound impact on the survival and recruitment of the species, which is already contending with multiple challenges. Consequently, understanding the contribution of offspring to population regeneration becomes vital, illuminating the mechanisms of population recovery and bolstering conservation efforts in a more informed manner.

The Japanese eel, *A. japonica*, is a catadromous species, with its spawning area located at approximately 142−143° E in the North Equatorial Current (NEC), west of the Mariana Ridge [8,9,10]. It is widely distributed in the western Pacific, ranging from the Philippines in the south, through Taiwan, mainland China, and Korea, to the north Japan [9,11,12,13,14]. After hatching, the eel larvae, referred to as leptocephali, are transported by the westward-flowing NEC and the northward-flowing Kuroshio towards the continental shelf, where they undergo metamorphosis into glass eels and eventually become pigmented elvers within estuaries [11]. Due to their complex life cycle, the development of artificial breeding technologies for this species has presented significant challenges, despite recent breakthroughs [15,16]. The procurement of the glass eels for aquaculture relies solely on the exploitation of wild resource, resulting in unsustainable fishing practices for the glass eels across Asian estuaries and coastal waters [6,11,12,17,18].

However, the intense and unrelenting fishing pressure has exacerbated the decline of the indigenous Japanese glass eel population. Wild populations have suffered a severe decline, with many populations vanishing from the majority of estuaries since the 1960s [18,19]. Around 2010, a worrying trend emerged: the annual recruitment plummeted by as much as 90% compared to the 1960s [17]. The Japanese eel’s status has led to its classification as “endangered” on the International Union for Conservation of Nature (IUCN) Red List of Threatened Species [20].

The Japanese eel, a delicacy in China, plays a significant role in the country’s aquaculture due to its consumer appeal and palatability. As a result of its popularity, China maintains a dominant position globally in the aquaculture and export of this species, accounting for approximately 70% of the world’s total output, or about 220,000 tons annually [21]. The Yangtze River estuary, a vital entry point for the glass eels entering their continental life phase, contributes to over 80% of China’s total glass eel catch. This substantial catch supports the development of *A. japonica* aquaculture, further solidifying China’s role in the international eel industry [7]. Since the early 1970s, the expansion of eel farming has led to overfishing the glass eels in the Yangtze River’s estuary, causing a marked decline in the counts of wild eel populations. To counterbalance this trend, starting from 1986, the government initiated a permit system at the regional level, strictly limiting the harvest of the glass eels in the Yangtze River’s estuary from January through April each year. However, despite these efforts, the population of the Japanese eels continues to decline [22]. To further protect and restore the Yangtze’s ecological health, from 1 January 2021, an all-encompassing fishing ban was implemented in the vital waters of the Yangtze River Basin, expected to last for a decade. Concurrently, on 20 November 2020, the Ministry of Agriculture and Rural Affairs established a specific non-fishing zone at the mouth of the Yangtze, aimed at preserving its aquatic ecology [23]. The ban measures enforced within this zone mirror those applied in the key waters of the Yangtze River Basin.

In the field of the Japanese eel conservation and assessment, accurately determining the recruitment rate of glass eels is a central concern. This task demands an in-depth understanding of eel population fluctuations across seasons and a detailed investigation of key factors affecting daily fishing efficiency. Moreover, the El Niño–Southern Oscillation (ENSO) events act as a major driver of interannual variability in the equatorial Pacific, significantly affecting the distribution, migration, and abundance of marine species, including the Japanese eel [24,25]. Recent studies have uncovered a significant correlation between El Niño phenomena and the decline in catches of glass eels of the Japanese eel, underscoring the importance of marine environmental changes in their recruitment [25,26]. It is hypothesized that the Japanese eel spawn in south of the salinity front of the North Equatorial Current (NEC), with larval transport within the NEC being integral to determining their abundance in East Asian waters. This salinity front is susceptible to positional shifts during ENSO events, which are crucial for the spawning migration of the Japanese eel [27]. In El Niño years, the northward bifurcation of the NEC into the Kuroshio Current diminishes, potentially impairing the efficiency of larval transport to their nursery habitats in East Asia [25,28]. Therefore, considering large-scale climatic phenomena such as El Niño is imperative in understanding migratory patterns and population dynamics to devise conservation strategies for the species, which are vital both ecologically and economically.

By focusing on the El Niño phenomenon and its potential impacts on the survival and recruitment of *Anguilla japonica*, a species already facing multiple challenges, we can integrate climatic factors into our analysis. This approach allows us to consider a broader array of influences, enabling a comprehensive assessment of the pressures faced by migratory fish species and enhancing our ability to predict population changes.

Previous research has primarily focused on the dynamics of the eel resources [22,29], their biological traits [30,31], and molecular biology [32] in the Yangtze River estuary, yet there has been a relative scarcity of long-term observations concerning the eel recruitment and its environmental drivers. Grounded in an eleven-year continuous survey from 2012 to 2022, the present study is designed to achieve several objectives: (1) determine the temporal pattern of the glass eel appearances; (2) analyze diurnal variations in catches and their association with environmental variables; (3) evaluate the specific impacts of the ENSO events on the eel recruitment; (4) identify and predict population trends and migratory patterns of glass eels of the Japanese eel. Through these endeavors, our principal goal is to deepen the understanding of recruitment patterns and influencing factors for the glass eels in the Yangtze River estuary, thus providing pivotal data for the sustainable management of the species and the conservation of the Yangtze River ecosystem.

## 2. Materials and Methods

### 2.1. Study Site

Our research was conducted at a strategically chosen location on the outer edge of the Yangtze River estuary in China (122°09′ E, 30°53′ N), located outside the no-fishing zone delineated by the Ministry of Agriculture and Rural Affairs. This site, historically a traditional fishing ground for glass eels of the Japanese eel, serves as a pivotal gateway for their entry into the Yangtze River. The site was selected due to being an easy to observe eel population, free from the distortive effects of fishing, thereby allowing for an accurate evaluation of the environmental factors that directly influence the natural recruitment patterns of the glass eels.

The ecological attributes of the site include an irregular semi-diurnal tidal regime, featuring two distinct tidal cycles per day with a range fluctuating from 1.5 to 5.2 m. The salinity, which experiences seasonal variations from 0 to 25 ‰, reflects the estuary’s diverse environmental conditions. Water depth often exceeding 10 m provides an unobstructed passage favorable for the glass eel migration. These characteristics make the site a suitable representative for the entirety of the estuary, offering a vantage point from which to gauge the natural recruitment processes endemic to this region.

### 2.2. Data Collection

#### 2.2.1. Glass Eel Catch Data

During the fishing seasons spanning from January to April across an eleven-year period from 2012 to 2022, catch data for the glass eels were collected using the same commercial boat authorized with fishing permits. This boat consistently gathered the glass eels from 400 eel nets, all of which were stationed at a fixed site (Figure 1). The eel nets, made from polyethylene thread, adopted a tapered configuration, with a gradual decrease in width from the mouth to the rear end. Each net was 16 m long and composed of two sections with different mesh sizes (3 mm and 1 mm) (Figure 2). The rear of the net mouth was connected to a floating seedling collection box, designed for the collection of the glass eels. Each net, featuring a rectangular mouth (4 m wide and 1.2 m high), was fastened to an expanded polystyrene float. This float was anchored to a fixed pile in the intertidal or subtidal zone with a 60 m rope. Net sampling was executed only at a near-surface depth of 0–3 m, as the upward buoyancy of the expanded polystyrene float allowed the net to move with the ebb and flow of the tidal currents. The orientation of the net was always towards the current to ensure consistent fishing efficiency during both ebb and flood tides. These eel nets are commonly used and are employed by nearly 95% of the licensed fishermen in the Yangtze River estuary.

Sampling was conducted over a single tidal cycle, lasting approximately 24 h and encompassing both a daytime and a nighttime flood tide. Subsequently, the total number of individual *A. japonica* glass eels captured was carefully recorded. The catch data were expressed in terms of the number of the glass eels captured per day by each boat, with each boat utilizing 400 nets.

#### 2.2.2. Environmental Parameters

Table 1 presented the relevant environmental parameters and their sources for the Yangtze River estuary. This estuary experiences an irregular semidiurnal tidal cycle, resulting in an average daily tidal range. This range is determined by calculating the average of the differences between the high and low tides over a day.

#### 2.2.3. Data Source to Distinguish the Phase of the El Niño–Southern Oscillation Events

El Niño and La Niña events were identified using data sourced from the NOAA’s Climate Prediction Center (URL: https://origin.cpc.ncep.noaa.gov/products/analysis_monitoring/ensostuff/ONI_v5.php, accessed on 1 December 2023). Our analysis incorporated monthly ENSO index values from 2012 to 2022, as derived from the Oceanic Niño Index (ONI). The ONI, which measures sea surface temperature anomalies in the El Niño 3.4 region (5° N–5° S, 120°–170° W), is calculated as a three-month moving average of the ERSSTv5 dataset. El Niño periods were defined as instances where ONI temperatures exceeded +0.5 °C for a minimum of five consecutive overlapping months, while La Niña periods were defined as those where ONI temperatures fell below −0.5 °C for the same duration. This categorization was chosen to analyze the impact of ENSO events on glass eel recruitment. In our study period (2012–2022), the years 2015, 2016, and 2019 were classified as El Niño years, while 2012, 2017, 2018, 2021, and 2022 were classified as La Niña years, and 2013, 2014, and 2020 were identified as normal years. To ensure a balanced analysis across different ENSO phases, biological sampling was strategically concentrated over a nine-year period from 2012 to 2020. This timeframe was carefully selected to include an equal number of El Niño, La Niña, and normal years.

#### 2.2.4. Biological Sampling

Biological specimens were systematically collected on the 20th of each month throughout the nine-year span from 2012 to 2020. Sampling dates were subject to adjustment in response to fluctuations in the fishing season, accommodating for instances such as a delayed start in January or an early end in April. The goal for each monthly collection was to gather at least 80 specimens to ensure robust biological analysis. Upon collection, each specimen was promptly measured for total length (TL, accurate to the nearest 1 mm) and wet body weight (BW, accurate to the nearest 1 mg). Subsequently, the pigmentation stage of each glass eel specimen was classified using the categories VA, VB, VIA1, VIA2, VIA3, VIA4, and VIB, as per the developmental criteria established by Fukuda et al. (2013) [34]. These stages describe the progression of pigmentation beginning with its first appearance at the tip of the caudal fin and nerve cord after metamorphosis (stage VA), and continuing through to stage VIB, where pigmentation appears uniformly across the entire body.

### 2.3. Descriptive Statistical Analysis

The total length, body weight, and daily catch data of the glass eels were subjected to Kruskal–Wallis tests to determine statistically significant variances across different months and ENSO events. Additionally, Spearman’s rank correlation analysis was employed to investigate the potential relationship between the peak recruitment dates of the glass eels and the interannual variability of the recruitment seasons [6].

### 2.4. Time Series Analysis

Continuous data are a prerequisite for time series analysis. However, challenges often arise during field data collection. For instance, our daily catch data contain gaps, primarily due to inclement weather making sampling infeasible. To mitigate this, we transformed all data into weekly time series by computing the weekly average of daily catches. This method effectively resolved the issue of missing daily data, thereby securing the data continuity required for time series analysis [35].

#### 2.4.1. Seasonal-Trend Decomposition of Time Series

Initially, we utilized the non-parametric Mann–Kendall (MK) test to examine the monotonic trends in the weekly average of daily catches. Subsequently, we applied the seasonal-trend decomposition using Loess (STL) technique to break down the weekly series of daily catch into its fundamental seasonal, trend, and remainder components [36]. The STL method deploys a continuous Loess line for smoothing the long-term component and 52-week specific Loess lines for the seasonal component. Each component undergoes iterative fitting until the resulting trend and seasonal components no longer deviate from those of previous iterations. The model includes two smoothing parameters representing the window widths of the seasonal and long-term components [36]. We selected window widths of 52 and 153 weeks, respectively, to visually reveal trends. Generally, window widths are chosen to produce relatively smooth long-term patterns that highlight potentially significant underlying changes while reducing the residuals to white noise. In our analysis, no discernible signals were found in the residuals, thereby directing our focus to the smooth, long-term, and seasonal patterns.

#### 2.4.2. Seasonal Auto-Regressive Integrated Moving Average Models

Auto-regressive integrated moving average (ARIMA) models are widely recognized as a standard approach in time-series forecasting. They function as a “filter” that separates the signal from the noise, subsequently projecting this signal into the future to generate forecasts. Central to ARIMA’s methodology is the assumption of stationarity within the response series. Achieving stationarity can be accomplished by differencing the series or by transforming the variable to stabilize the variance or mean [35].

Given the distinctive seasonal and nonlinear regression attributes characterizing the daily catch of Japanese glass eels, we developed a seasonal ARIMA model by incorporating additional seasonal terms. The seasonal ARIMA model is often expressed as ARIMA(p, d, q)(P, D, Q)_m_, where “p”, “d”, and “q” stand for the orders of autoregression (AR), differencing, and moving average (MA), respectively. Concurrently, “P”, “D”, and “Q” denote the seasonal orders of AR, differencing, and MA correspondingly, while “m” signifies the length of the seasonal period [37].

In the present study, we opted for a weekly time series with a seasonal period “m” of 52. The series underwent differencing until stationarity was achieved. Suitable AR and MA orders were then chosen, informed by the autocorrelation function (ACF) and the partial autocorrelation function (PACF). The lowest Akaike’s information criterion (AIC) value guided the selection of the optimal forecasting model. Following model selection, we applied the Ljung–Box test to verify the model’s assumptions, ensuring that the residuals were random and lacked autocorrelation, thus confirming the model’s validity.

### 2.5. Generalized Additive Model Analysis

To elucidate the environmental determinants of daily catch for *A. japonica*, we applied generalized additive models (GAMs). These models excel at revealing complex non-linear relationships through the use of smoothing functions, providing deeper insights than those afforded by linear models or simple correlation analyses [38]. Our GAM featured a Gaussian error distribution with an identity link function, modeling the natural logarithm of the catch data. The model encompassed both nominal (fishing season and the annual ENSO phase) and continuous (SST, tidal range, and lunar phase) explanatory variables.

The GAM was structured as follows:Log (daily catch) ~ a + s(SST) + s(tidal range) + s(lunar phase) + ENSO phase + fishing season(1)
where s() represented a smoothing cubic spline and a represented the intercept. Prior to modeling, we assessed multicollinearity among predictors using variance inflation factors (VIFs) [39], excluding any predictors with a VIF exceeding 3 [40]. Model fit was ascertained through the proportion of the total deviance explained and AIC. We employed a forward selection process, commencing with a null model and sequentially adding variables that mostly enhanced the explanatory power and reduced AIC. Only those variables that conferred no significant improvement were omitted from the final model. Diagnostic assessments, including analyses of residual histograms and plots of residuals against fitted values, were performed to ensure the model’s fit and to verify that the assumptions of normality and homogeneity of variance were met.

All statistical analysis, modeling, and plotting were conducted using R version 4.3.1 (http://www.r-project.org), which was accessed on 5 February 2023, employing the “mgcv” [41], “dplyr” [42],”trend” [43],”stats” [44], and “ggplot2” packages [45].

## 3. Results

### 3.1. Biological Characteristics and Pigmentation Stages

All the collected eels were identified in the glass eel stage, with pigmentation stages ranging from VA to VIA1. Specifically, stage VA is characterized by the initial appearance of pigmentation at the tip of the caudal fin and nerve cord. In contrast, stage VIA1 is marked by pigmentation extending along the dorsum from the tip of the caudal fin to just anterior of the dorsal fin. The most prevalent stage was VB1, comprising 65.6% of the total, followed by VA at 27.8%. The glass eels in stages VA and VB1 were considered new recruits in the estuary. Other stages were less represented, with VB2, VIA0, and VIA1 accounting for 5.5%, 0.9%, and 0.3% of the sample, respectively. As the fishing season progressed, there was a slight decline in the proportion of early-stage eels and a corresponding increase in later stages, indicating a clear seasonal trend (Figure 3).

TL of the glass eels varied from 47 mm to 63 mm, with an average of 56 ± 2 mm (standard deviation), while BW ranged between 56 g and 214 g, yielding an average of 119 ± 26 g. The most frequent TL and BW groups were 54–57 mm and 100–140 g, accounting for 44.6% and 55.0% of the total, respectively (Figure 4).

Significant differences were observed in both TL and BW among different pigmentation phases (Kruskal–Wallis test, all *p*-values < 0.05). Specifically, there was a significant decreasing trend in both TL and BW as pigmentation advanced, with a more pronounced decreasing trend observed for BW (Figure 5). Comparison of TL and BW of the newly recruited glass eels (stages VA and VB1) across different years and months revealed no significant differences in TL (Kruskal–Wallis test, *p* = 0.20 and *p* = 0.16), However, there were significant yearly and monthly variations in BW (Kruskal–Wallis test, all *p*-values < 0.05), which showed a notable seasonal decrease and was lower in La Niña years (108 ± 22 g) compared to normal (121 ± 24 g) and El Niño years (130 ± 25 g) (Appendix A).

### 3.2. Daily Catch and Recruitment Patterns

The time series data of daily catch over 11 consecutive fishing seasons revealed obvious fluctuations in daily catch, with the average daily catch ranging from 153 individuals in 2012 to 1108 individuals in 2022 (Figure 6). There were significant differences in daily catch among the different years (Kruskal–Wallis test, *p* < 0.001).

Additionally, the analysis of arrival patterns revealed that the earliest first capture occurred on 6 January 2012, while the latest first capture was observed on 29 January 2013. Arrival peaks were observed throughout all lunar phases, with the majority (87.9%) occurring during the full moon (lunar days 12–18) and new moon (lunar days 27–3), with frequencies of 15 and 14 times, respectively. Only a minority (12.1%) of the arrival peaks were observed during the first quarter moon (lunar days 4–11) and last quarter moon (lunar days 19–26), with frequencies of 1 and 3 times, respectively (Figure 6). The intervals between peak periods ranged from 14 to 45 days, with an average interval of 24 ± 11 days.

Furthermore, when analyzing the average daily catch during normal, La Niña, and El Niño years, it was found that the values were (618 ± 677), (596 ± 593), and (421 ± 263), respectively. While the mean daily catch was lowest during El Niño years, statistical analysis revealed no significant differences in catch abundance among the three groups (*p* = 0.15). Comparing the monthly catch data between El Niño and La Niña years, it was observed that the recruitment season tended to commence earlier during El Niño years (Figure 7).

### 3.3. Seasonality, Trend, and Forecasting in Weekly Average Daily Catch

The analysis showed a significantly strong increasing trend (MK, s = 126,270, *p* < 0.001) of daily catch of the glass eels from 2012 to 2022. An STL model analysis further explored the long-term and seasonal trends of the average daily catch of the glass eels on a weekly basis. The long-term trend showcased a three-year cycle, most notably during surge periods from 2012 to 2014, 2015 to 2017, and 2018 to 2020, each subsequently followed by a downturn. However, there was a consistent increase in the annual catch after 2020, altering the previous triennial oscillation. Besides the long-term trend, the seasonal trend revealed a bimodal catch pattern, suggesting two significant migration waves of the glass eels each recruitment season, with an interval of approximately five weeks between them (Figure 8).

The ARIMA(4,0,0)(0,1,1)_52_ model was chosen for its minimal AIC value. The residuals, confirmed as white noise by the Ljung–Box test (*p* = 0.77) and evidenced by the ACF and PACF plots, demonstrated the model’s validity. The histogram of forecast errors demonstrated a random pattern with fluctuations around zero, reflecting the model’s efficacy (Appendix A). The model’s forecast for the upcoming two years (2023–2024), as shown in Figure 9, anticipates a consistently high level of annual glass eel catches throughout the fishing seasons of 2023 and 2024. The daily catch averages are projected to be 969 and 1008 eels, respectively. The seasonal trend is predicted to maintain a bimodal pattern, with the peak catch days falling on 17 April 2023, and 30 March 2024 (Figure 10). A Spearman correlation analysis indicated a continuing trend of seasonal delay in these peak catch dates, extending from 2016 through 2024 (*r* = 0.714, *p* = 0.03).

### 3.4. Generalized Additive Model Analysis of Factors Influencing Daily Catch

Given the smallest AIC value, Model 4 was determined as the preferred model for this study. The AIC score for this model was 2082.724, and it accounted for a total variance of 47.7% (Table 2). Upon assessing the diagnostic plots of the residuals in the final model (Appendix A), it was observed that the model’s assumptions of the model, including the normal distribution and homoscedasticity of the residuals, were not violated. Though the final GAM model revealed significant influences of the fishing season (*p* < 0.001), tidal range (*p* = 0.04), water temperature (*p* = 0.01), and lunar phase (*p* = 0.02) on the daily catch, the fishing season emerged as the dominant factor affecting recruitment at this estuary. The fishing season accounted for the greatest impact (82.2%) on the daily catch of *A. japonica* glass eels in this model. Furthermore, other environmental variables such as water temperature (11.9%), lunar phase (4.4%), and tidal range (1.5%) also exhibited significant effects on the daily catch, albeit to a lesser degree (Table 3).

According to the final model (Figure 11), positive values on the *y*-axis indicate a significant impact on the daily catch. In terms of interannual effects, the model revealed that higher daily catches were consistently observed in the years 2014, 2017, and 2020–2012 compared to other years. Regarding the environmental factors, the study found that the optimal water temperature range for achieving a larger catch of *A. japonica* glass eels is between 6 °C and 8 °C, as well as 10–12 °C. Regarding the impact of lunar phase and tidal range, a consistent pattern emerged, showing that higher catches were correlated with a tidal range of 4–5 m (at high tide) and during periods of new or full moon.

## 4. Discussion

### 4.1. Recruitment of Japanese Eel in the Yangtze River Estuary

This study analyzed the daily catch variability of *A. japonica* glass eels in the Yangtze River estuary, covering 11 consecutive recruitment seasons annually from 2012 to 2022. The recruitment season in the Yangtze estuary generally begins in January, while it starts in February in the northern regions of Republic of Korea [46]. Conversely, the season commences even earlier, in November and December, in the southern regions such as Taiwan and the Pearl River estuary in China [6,13]. The migration of *A. japonica* larvae to the East Asian Continental Shelf is a passive process influenced by the North Equatorial Current, the Kuroshio, and various regional coastal currents. The timing of larval arrival at different East Asian locales is dictated by these currents and the respective distances from the spawning grounds. Our findings indicate that the complex interplay of currents and the cooler coastal waters can hinder larval migration, potentially precipitating earlier arrivals in warmer, lower latitude areas or those closer to the Kuroshio Current’s influence.

In contrast to the stable yet low recruitment of Japanese eel at the Pearl River estuary [6], the daily catch of the glass eels in the Yangtze River estuary exhibited significant intra-annual and interannual variability. STL analysis confirmed a triennial pattern in the glass eel catches at the Yangtze estuary, corroborating long-term trends highlighted by previous research. Furthermore, it uncovered a bi-modal seasonal pattern, suggesting the presence of two distinct migration waves within each recruitment season. It is likely that this pattern of glass eel recruitment is a result of mature eels spawning during the new moon phases [47]. The proposed mechanism of new moon spawning is believed to trigger the arrival of the glass eels in batch-like waves, adhering to a monthly rhythm. The observed five-week interval between these arrival peaks aligns with the lunar cycle, lending support to the hypothesis that lunar phases play a significant role in influencing the spawning and subsequent recruitment of eels.

While the initial analysis emphasizes environmental factors affecting the glass eel recruitment, addressing the status of the parental eel stock and the impact of fisheries activities is equally essential. The health and abundance of the parental eels are fundamental to maintaining consistent recruitment levels. A notable trend is the sharp decline in the number of mature eels in the Yangtze River, largely due to overfishing and environmental changes [48]. This reduction in the breeding population could negatively impact the recruitment process of glass eels. Moreover, specific fishing practices not only threaten the juveniles but also the adult eels, potentially reducing recruitment success rates. Furthermore, habitat destruction, such as agricultural lake enclosures, disruptions in river–lake connections, and pollution, also adversely affects the spawning of parental eels and the survival of their offsprings [49].

Several studies have suggested that interannual variability in the recruitment of the glass eels may be connected to changes in climate regimes, particularly events associated with the ENSO [24,25,27,50]. These events are known to influence the transport and bifurcation latitude of the North Equatorial Current (NEC), which may lead to nutrient-depleted marine environments or significantly prolonged migration times during the leptocephalus stage of eels, ultimately affecting the recruitment success of glass eels. A statistical analysis of glass eel catch data in Japan demonstrated that during El Niño years, the number of the Japanese eel larvae transported to the Kuroshio Current was significantly lower than those transported to the Mindanao Current, which corresponded with decreased recruitment. Conversely, in non-El Niño years, the number of larvae transported to the Kuroshio was twice as high as in El Niño years, resulting in increased recruitment [25]. Our study indicated that the average daily catch of the glass eels in the Yangtze estuary during El Niño years, though lower than in normal or La Niña years, did not significantly differ. Furthermore, GAM analysis showed that the impact of ENSO on the daily catch of the glass eels was not significant. These findings imply that ENSO may not be the primary influence on the yearly variability in glass eel recruitment in the Yangtze estuary. Conflicting results may have stemmed from variations in the disparate pathways of ocean currents affecting the recruitment of the glass eels from their spawning grounds to East Asia. Tracer simulation tests have shown that after departing from their spawning grounds, the larvae of the Japanese eel are directly transported to the continental shelves of Japan via the main stream of the Kuroshio Current. In contrast, the pathways facilitating the transport of larvae to the Yangtze River estuary were comparatively more intricate [51]. The migration of Japanese eel larvae to the Yangtze River estuary was predominantly conducted along pathways associated with the Kuroshio Current and the Yellow Sea Warm Current (YSWC). The western branch of the YSWC extended eastward near the Zhejiang waters, merging with the southward-flowing Yellow Sea Coastal Current (YSCC) and giving rise to a minor counterclockwise gyre. This gyre likely played a significant role in facilitating the transport of the glass eels to the Yangtze River estuary via a sequential YSWC and YSCC pathway. Also, there might be another migratory pathway through the Taiwan Strait Warm Current (TSWC) aiding in the dispersal of the glass eels from northern Taiwan to the Yangtze River estuary [51]. The complex coastal currents may obscure the impact of El Niño events on the recruitment waves of the glass eels in the Yangtze River estuary.

Moreover, during El Niño years, the recruitment of the glass eels in the Yangtze estuary tends to commence earlier, a trend that is particularly evident in years with strong El Niño events. For instance, Liu et al. observed that in the strong El Niño year of 1998, the recruitment of the glass eels during January and February constituted 20.90% and 54.6% of the annual total, respectively. In contrast, in the following La Niña year of 1999, the proportions were much lower in these months, at 4.28% and 36.85%, respectively [22]. This pattern indicates that El Niño events can indeed exert an influence on the timing of glass eel recruitment.

It is noteworthy that ARIMA models forecast glass eel recruitment patterns extending into the next two years (2023–2024) and have also detected a shift toward later recruitment seasons beginning in 2016. These shifts could plausibly be interpreted as an adaptive response to environmental fluctuations, especially in view of a notable warming trend in SST in the Yangtze estuary, which have increased by an average of 0.48 °C per decade over the past forty years [52]. Increased sea surface temperatures may adversely influence glass eel recruitment. Nonetheless, there is potential for the glass eels to mitigate these negative effects by adjusting their migratory timing to later in the season. Our study also identified a significant decrease in the average body weight of the glass eels arriving later, indicating that such late recruits may lack adequate energy reserves for subsequent upstream migration. Consequently, a shift toward later recruitment seasons may have detrimental effects on the population dynamics, underscoring the imperative for sustained and detailed long-term monitoring and assessment.

### 4.2. The Influence of Local Environmental Factors

Numerous studies have established that the recruitment dynamics of the glass eels are influenced by a variety of environmental factors such as the lunar phase, tidal cycle, diurnal rhythm, moonlight, water temperature, salinity, turbidity, olfactory cues, and rainfall [12,19,46,53,54,55,56,57,58,59], yet discrepancies in findings are noted [58].

Water temperature has been identified as a critical factor affecting the inshore migration of eels, a finding that is supported by extensive research [6,12,46,56]. Our study revealed that the timing of the glass eel migration from the ocean to estuaries seemed to be triggered by water temperature, with an optimal range that facilitates migration in the Yangtze River estuary. The glass eels of the Japanese eel were observed to migrate most effectively within temperature intervals from 6 °C to 8 °C and 10 °C to 12 °C, coinciding with significant peaks in daily catches indicative of major migration waves.

Drawing from Han et al. (2011), we postulated that at temperatures below 5 °C, the glass eels may experience prolonged starvation, weight loss, and delayed pigment development [60]. A marked increase in the glass eel captures was recorded as water temperatures rose above 5.5 °C, particularly within the range from 6 °C to 8 °C during winter, consistent with patterns described in previous studies of the Geum River and Pearl River estuaries [6,46]. The subsequent large-scale migration wave, typically prompted by rising water temperatures, is thought to be linked to their upstream migratory behaviors. Experimental evidence suggested a strong positive correlation between migratory activities of the glass eels and water temperatures above a certain threshold [61]. Nonetheless, we observed a notable decline in glass eel abundance as water temperatures exceeded 13 °C, with only 5.2% captured in waters above 14 °C over eleven fishing seasons, likely due to faster developmental rates at higher temperatures precipitating early migration and settlement. Interestingly, the second migration wave, associated with higher temperature thresholds, has not been documented in the Geum River or Pearl River estuaries. The distinct bimodal recruitment pattern in the Yangtze River estuary, particularly the pronounced peak in the later recruitment phase, might be a contributing factor to the relatively higher abundance of the glass eels in this region compared to other estuaries with lower recruitment levels.

Apart from water temperature, our observations indicate that lunar phases and tidal ranges notably influenced the daily catches. Specifically, a mild positive correlation had been documented between these two environmental factors and eel abundance, with peaks frequently coinciding with spring tides during full or new moons. The GAM also suggested that the migration of the glass eels displayed semi-monthly fluctuations, with nearly identical likelihoods of peaks during new moon or full moon phases. This semilunar periodicity in glass eel catches, driven by the coupled relationship of the moon and tides, had been reported in European eels [62], American eels [63], and other members of the genus Anguilla, including the short-finned eel and the Australian eel [58,64,65]. Such semilunar periodicity in the catch was considered to reflect the glass eels’ behavioral response to tidal amplitude [53,64], with Anguillid glass eels exhibiting tendencies to move with the tides, particularly showing more activity during rising tides [66,67,68]. This behavior was likely an adaptation to the substantial hydrodynamic forces during spring tides, allowing them to utilize the stronger currents to move further inland or upstream. The powerful flow during spring tides not only increased the interception rate of fishing nets but was particularly advantageous for those weaker swimming glass eels to be swept into the nets from tidal flats. The facilitative role of tidal forces was critical for glass eels; it prompted these juveniles to adapt to tidal rhythms by adjusting their swimming direction and depth—a key step in their crucial migration from marine to freshwater habitats.

However, our study also noted cases where high eel abundance did not coincide with extensive tidal ranges. For instance, in 2014 and 2022, peaks in the eel catch were observed even during neap tides. Tsukamoto et al. (2003) collected *A. japonica* leptocephali in July 1991 near a spawning area, which included individuals hatched during the new moon periods of May and June. They reported mixing of monthly cohorts even near the spawning site [47]. Drawing from this, we propose that the glass eels arriving in the Yangtze River estuary may not only come from larvae spawned in the 12–16° N region during the new moon but also from the mixing of monthly cohorts. These cohorts may have followed slightly different paths on the NEC and Kuroshio currents to the estuary. This may lead to a continuous recruitment pattern, potentially explaining the sustained high catches observed during both spring tides and neap tides over periods from 20 to 30 days encompassing multiple tidal cycles.

Additionally, the influence of tidal range on the glass eel catch appeared to be less pronounced at the start and toward the end of the fishing season. In the early phase, the arrival of fewer individual recruits might obscure the effect of tidal ranges. On the other hand, during the late phase of the season, a greater number of older pigmented glass eels with stronger swimming abilities against the current could reduce the impact of tides on the catch.

## 5. Conclusions

Our study has illuminated the recruitment patterns of *Anguilla japonica* in the Yangtze River estuary and their interplay with local environmental dynamics. While we have made strides, substantial uncertainties still linger regarding the drivers of recruitment variability, underscoring the complexity inherent in the eel population studies. This complexity underlines the need for strategic planning within the fishing season.

To refine fishing season planning, we propose that future management strategies encompass:

Optimal timing of fishing efforts: An investigation into the correlation between environmental factors—such as salinity, water temperature, and lunar cycles—and daily catch rates enables fisheries managers to pinpoint peak recruitment periods within the January to April fishing window. During these critical junctures, the stricter regulations should be enforced to moderate fishing intensity, balanced by more lenient measures in times of decreased recruitment, thus safeguarding against overfishing when the juvenile eels are most vulnerable.

Adjustments based on environmental indicators: Our findings suggest a strong correlation between specific environmental conditions and heightened recruitment rates. Consequently, fishing efforts should be dynamically calibrated in response to these indicators, optimizing catch efficiency and mitigating impact on the eel population.

Integrated policy and adaptive management: The current ten-year fishing moratorium in the Yangtze River Basin presents a unique chance to embed these strategic planning elements into broader conservation efforts. Careful monitoring during this period is imperative to evaluate its impact on the eel demographics. If preliminary observations are any indication, we may already be witnessing shifts in the eel population dynamics, which will inform adaptive management strategies following the ban’s conclusions.

Future research will aim to broaden the dataset, capturing more detailed information on spawning population sizes, hatchling numbers, and key environmental metrics in spawning zones, such as ocean currents and chlorophyll content. Furthermore, the narrow scope of our study calls for enhanced spatial and longitudinal monitoring to unravel long-term and cyclical patterns in the eel recruitment. As the Yangtze’s ten-year fishing prohibition continues, steadfast monitoring is essential to determine its influence on the resurgence of the eel population and the health of the ecosystem. The insights gleaned will prove crucial in gauging the moratorium’s effectiveness and shaping future resource management and conservation policies, ensuring that strategic fishing season planning is synergistic with overarching conservation goals.

## Figures and Tables

**Figure 1 biology-13-00056-f001:**
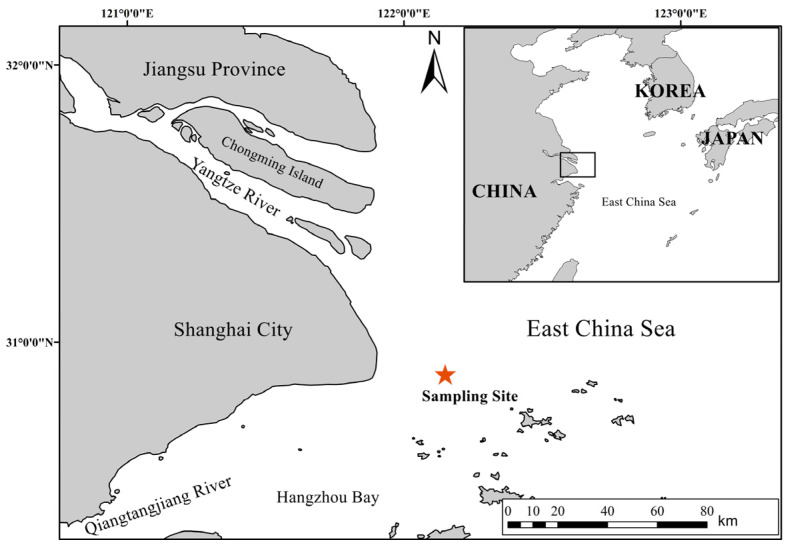
Geographic location of the *Anguilla japonica* glass eel collection site in the Yangtze River estuary.

**Figure 2 biology-13-00056-f002:**
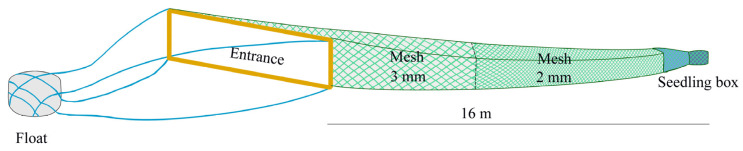
Schematic diagram of the eel net employed for capturing *Anguilla japonica* glass eels in the Yangtze estuary.

**Figure 3 biology-13-00056-f003:**
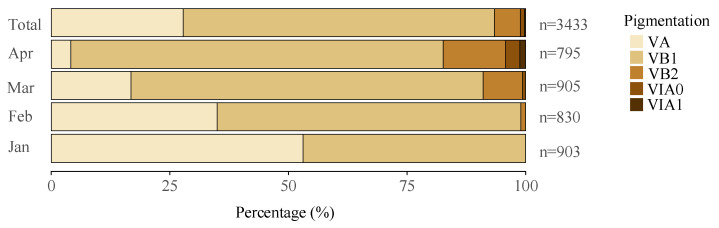
Samples were collected over the years 2012–2020, with a total of 3433 specimens analyzed. Proportional distribution of the glass eels by pigmentation stages in the Yangtze River estuary, showing both overall percentage composition and monthly variations in stage distribution. Data collected from 2012 to 2020 analyzed a total of 3433 specimens. “n” represents the total number of specimens for that month or the total count.

**Figure 4 biology-13-00056-f004:**
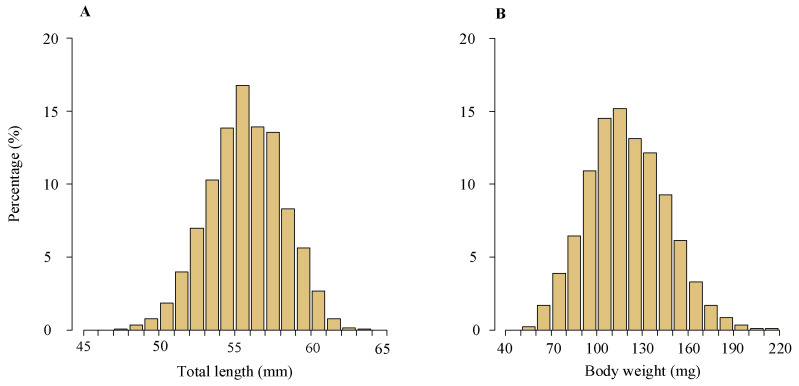
Length and weight frequency distributions of the glass eels in the Yangtze River estuary. This includes length frequency distribution (**A**) and weight frequency distribution (**B**) of the glass eels collected from the area. The dataset spans 2012–2020 and includes analysis of 3433 specimens.

**Figure 5 biology-13-00056-f005:**
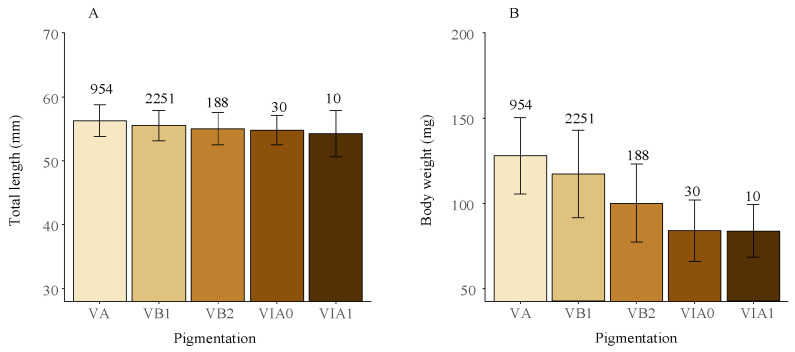
Average total length (**A**) and body weight (**B**) of the glass eels in the Yangtze River estuary across different pigmentation stages, with standard deviations shown as error bars. Data from 2012 to 2020, covering 3433 specimens, were included. Counts for each pigmentation stage are indicated in the figure.

**Figure 6 biology-13-00056-f006:**
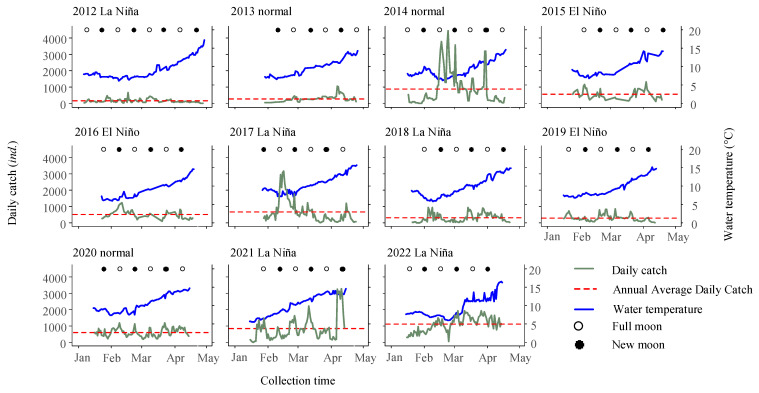
Daily fluctuations in glass eel catch (individuals per boat per day) and water temperature are observed during the fishing periods spanning from 1 January to 30 April each year, from 2012 to 2022, in the Yangtze River estuary in China. The water temperature trend is clearly indicated by a pronounced blue line, while the deep green line marks the daily catch rates. The lunar phases, including new and full moons, are represented by solid and hollow circles. Furthermore, a distinctive red dashed line accentuates daily catch for the entire fishing season.

**Figure 7 biology-13-00056-f007:**
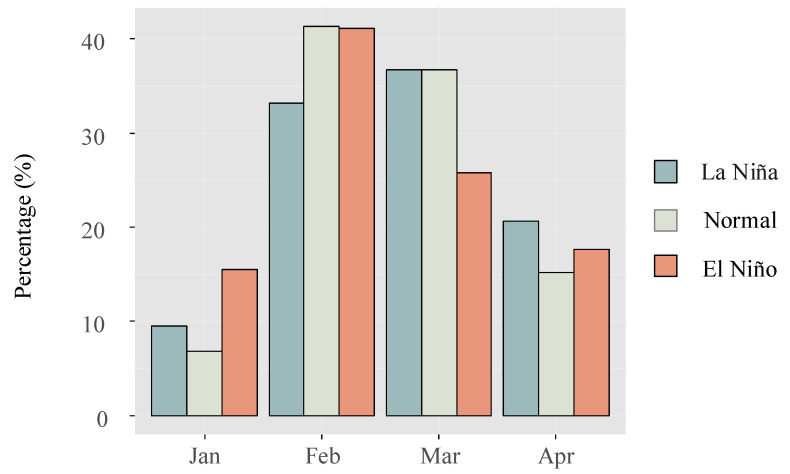
Comparison of monthly catch percentages for the glass eels in the Yangtze River estuary during distinct ENSO events.

**Figure 8 biology-13-00056-f008:**
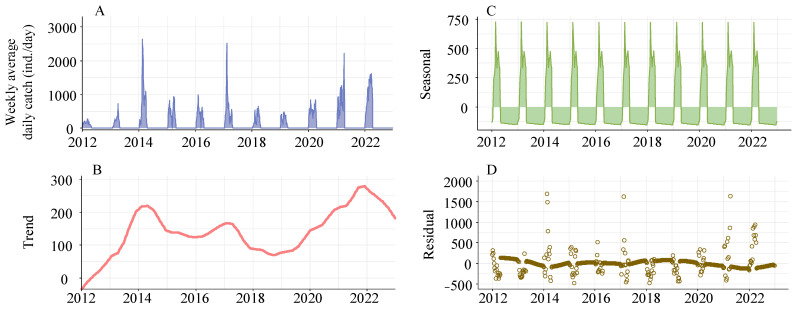
Decomposition of seasonal trends in *Anguilla japonica* glass eel catches in the Yangtze River estuary using Loess (STL). Panel (**A**) displays the actual weekly average daily catches. Panel (**B**) illustrates the extracted long-term trend component. Panel (**C**) depicts the seasonal cycle component. Panel (**D**) presents the remaining residuals after trend and seasonal components have been removed.

**Figure 9 biology-13-00056-f009:**
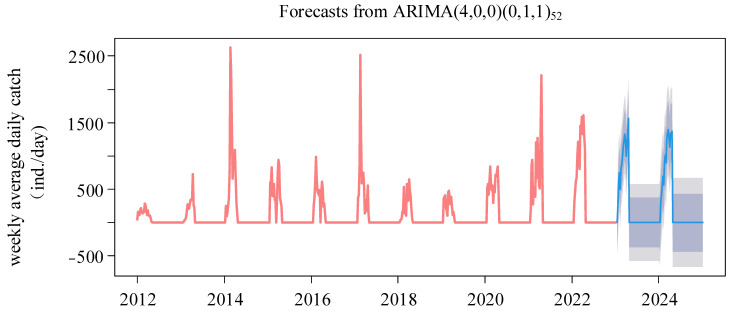
Forecasts of weekly average daily catch time series of the glass eels from 2012 to 2024. The bright blue line is the forecast, and the dark gray area and the light gray area are the 80% and 95% confidence levels, respectively.

**Figure 10 biology-13-00056-f010:**
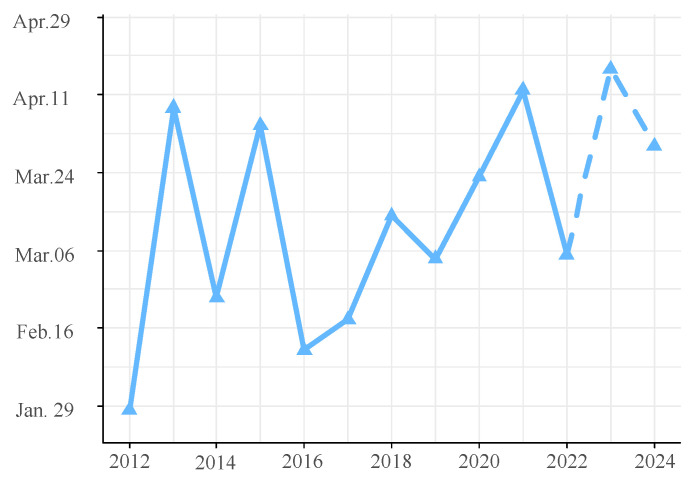
Dates of maximum glass eel catch in the Yangtze River estuary. The solid line marks the actual dates with the highest catches, while the dotted line represents the predicted peak catch dates.

**Figure 11 biology-13-00056-f011:**
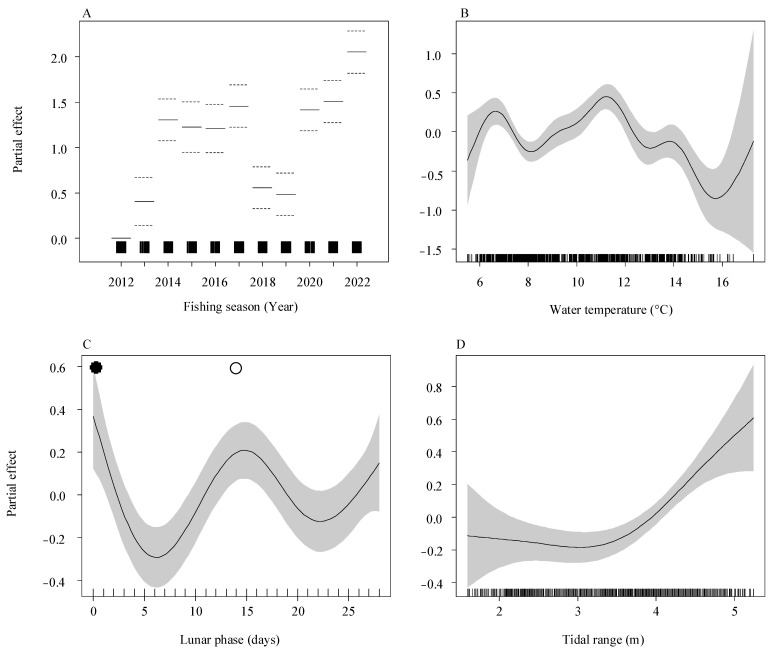
Partial effects of environmental variables on the daily catch of *Anguilla japonica* glass eels as determined by the final generalized additive model (GAM). The relationships are shown for (**A**) fishing season, (**B**) water temperature, (**C**) lunar phase, and (**D**) tidal range. The *y*-axis indicates the estimated partial effect size, with shaded areas representing the 95% confidence intervals. Rug plots show the data point density along the *x*-axis. Dashed lines in (**A**) mark the 95% confidence intervals. In (**C**), new and full moons are indicated by filled and open circles, respectively.

**Table 1 biology-13-00056-t001:** Environmental variables, their descriptions, and data sources.

Variable Name	Description	Source
water temperature	daily sea surface temperature (°C) (121°875′ E, 30°625′ N–122°375′ E, 30°375′ N)	NEAR-GOOS regional delayed mode database (RDMDB, accessed on 1 December 2022)
tidal range	difference in water height between low and high tide (m)	China Oceanic Information Network (https://www.nmdis.org.cn/, accessed on 1 December 2022)
lunar phase	days elapsed since the last new moon, which corresponds to the lunar cycle’s duration of approximately 28 days. This measure captures the complete lunar phase progression, from new moon to full moon and back, and includes the occurrence of neap and spring tides as influenced by the lunar cycle. The methodology for quantifying lunar phases follows the approach used by Fukuda et al. (2016) [33].	China Oceanic Information Network (https://www.nmdis.org.cn/, accessed on 1 December 2022)

**Table 2 biology-13-00056-t002:** Selection of generalized additive models for predicting glass eel daily catch, considering fishing season, ENSO phase, SST, tidal range, and lunar phase. Models ranked by AIC, with Model 4 (bolded) as the chosen model, showing ΔAIC for incremental variable inclusion.

Model	Deviance Explained (%)	AIC	ΔAIC
Null		2618.798	434.553
Mod1: Fishing season	39.2	2184.245	72.900
Mod2: Mod1 + SST	44.9	2111.345	25.512
Mod3: Mod2 + lunar phase	47.0	2085.833	3.109
**Mod4: Mod3 + tidal range**	**47.7**	**2082.724**	**0**
Mod5: Mod4 + ENSO phase	47.7	2082.725	−0.001

**Table 3 biology-13-00056-t003:** Contributions of environmental factors in the final generalized additive model for glass eel daily catch. Factors include fishing season, sea surface temperature, tidal range, and lunar phase. “edf” represents estimated degrees of freedom, *p*-values denote significance levels, and “total deviance explained” shows the proportion of the model’s variance accounted for by each variable.

Variable	edf	*p*	Deviance Explained (%)
Fishing season	10	<0.001	82.2
SST	8.595	0.01	11.9
Lunar phase	5.511	0.02	4.4
Tidal range	1.622	0.04	1.5

## Data Availability

The data supporting the findings of this study are available from the College of Fisheries and Life Science, Shanghai Ocean University (contact: Hongyi Guo; email: hy-guo@shou.edu.cn).

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
