# Peer review of "Recruitment Patterns and Environmental Sensitivity of Glass Eels of Anguilla japonica in the Yangtze Estuary, China"

_biology, 2024, doi:10.3390/biology13010056_

Round 1

Reviewer 1 Report

Comments and Suggestions for Authors

Comments on Biology-2815150: Temporal Patterns and Environmental Influences on Anguilla japonica Recruitment in the Yangtze Estuary

The study works on the comprehensive data set of the number of glass eels of Anguilla japonica in the Yangtze estuary and then link to environmental factors, i.e. water temperature, tidal range and lunar cycle to forecast the future recruitment. The prime mention is that should the fishing intensity be incorporate to the study, either by as a factor that govern the recruitment or proofing that this factor is minimal effect to the recruitments in the past studied years. It is generally accepted on the fisheries is among the main factor effecting both broodstocks and the glass eels themselves. Also are there any extreme weather events, e.g. storm surge, taken into the calculation?

Some specific comments are as followed

Simple summary: Why the temperature is not mentioned as 6-8 oC and 10-12 oC? Is this because of the different life stages of the glass eels?

Abstract: The numeric results must be added into this section. Abbreviation, i.e., ARIMA must be motioned in full name as not all readers are familiar with this term.

Introduction:

·        Lines 86-88: This sentence is required reference(s).

·        Line 93: The sentence is not clear, since the fishing season itself is quite limited, i.e. January to April, how the season could be strategically planned? It is not clear about this point in either Discussion or Conclusion

Materials and Methods:

·        Better detail the sampling site on how is it considered as the representative site for the whole estuary.

·        Better revise the description of lunar phase in Table 1

·        References are required for sub-sections 2.3, 2.4, 2.4.1, 2.4.2 and 2.5.

·        Full name has to be presented at its first mentioned. Not all readers are familiar to the terms used in the manuscript.

·        Every R packages must be cited properly.

Results:

·        What are VA to VIA1 both in text and Tables? They are never prior mentioned. Detail of these stages must be elaborated.

·        Make consistency in the digit number of each result.

·        If the P-value is expressed as exact value, it is not necessary to add < 0.05 or > 0.05.

·        What is the title for horizonal axis in Table 6.

·        Revise Tables 2 and 3 to make them easier to be understood as well as check the abbreviations.

·        Why Figure 11(a) is not present as “line” graphs as other figures, i.e. 11(b) to 11(d) 

Discussion: Su-section 4.1 should discuss also on the status of the parental stocks as well as the effects by fisheries.

END OF REVIEW

Comments on the Quality of English Language

Minor editing is required.

Author Response

Dear Reviewer,

We sincerely appreciate your thorough review and insightful suggestions on our manuscript, "Temporal Patterns and Environmental Influences on Anguilla japonica Recruitment in the Yangtze Estuary." We have carefully considered each of your comments and undertaken substantial revisions to enhance our paper. For your convenience, these amendments are marked in red within the manuscript for easy identification and review. Below, we outline our responses to each of your key points:

Regarding the Consideration of Fishing Intensity as a Research Factor:

   - To ensure minimal influence from previous fishing activities, we conducted our sampling at the Yangtze Estuary's outermost points. This strategy guarantees that our data predominantly reflect the direct impact of environmental factors on the natural recruitment patterns of glass eels.

   - Over the 11-year study period, the consistency in our fishing methods—using identical vessels and standardized gear—ensures the reliability and comparability of our long-term data.

   - Recognizing the importance of numerical precision, we have tailored the decimal places for each data type such as total length, weight, percentages, p-values, and AIC values, based on their respective statistical standards and measurement precision. This ensures uniformity in our reporting and reflects our commitment to providing data that is both accurate and practically useful, while maintaining clarity and coherence.

On the Impact of Extreme Weather Events:

   - Due to safety concerns, our fishing activities are suspended during extreme weather conditions. This is a limitation in our current dataset. In future studies, we intend to include data on extreme weather events for a more comprehensive analysis of environmental impacts.

Responses to Specific Comments:

  •  Abstract Simplification: We have refined the abstract to highlight the substantial influence of water temperature on glass eel migration, particularly focusing on the critical temperature ranges of 6-8°C and 10-12°C and their correlation with migration patterns. This clarification enhances the ecological significance of our findings for conservation efforts.
  •  Abstract: The abstract now includes specific numerical results. We have also ensured that all first-time abbreviations, such as ARIMA (Auto Regressive Integrated Moving Average), are accompanied by their full terms.
  •   Introduction:

     1) We have addressed the issue of missing citations in lines 86-88 by adding relevant references.

     2) We have elaborated on strategic planning for limited fishing seasons in our revised conclusion, highlighting the importance of synchronizing fishing activities with environmental variables and recruitment rates.

  •   Materials and Methods:

     1) We have clarified how our sampling sites representatively cover the entire Yangtze Estuary.

     2) The description of lunar phases in Table 1 has been revised for clarity, following the methodology of Fukuda et al. (2016).

     3) Per your request, we have added necessary references in sections 2.3, 2.4, 2.4.1, 2.4.2, and 2.5. All abbreviations are now fully defined at their first mention, and all R packages used are correctly cited in the reference list.

  •  Modifications to the Results Section:

     1) We have enriched the information for stages VA to VIA1 and maintained consistency in numerical precision.

     2) Recognizing the importance of numerical precision, we have tailored the decimal places for each data type such as total length, weight, percentages, p-values, and AIC values, based on their respective statistical standards and measurement precision. This ensures uniformity in our reporting and reflects our commitment to providing data that is both accurate and practically useful, while maintaining clarity and coherence.

     3) Clarification on P-Value Representation:

        - For extremely small P-values, we use notations like '<0.05'. We have corrected other P-values for precise representation as required.

  •    Modifications to Tables and Figures:

     1) Misunderstandings related to 'Table 6' have been corrected, and 'Figure 6' now clearly indicates that the horizontal axis represents 'collection time'.

     2) We acknowledge and thank you for your detailed review of 'Figure 11', particularly Panel A. In response, we have clarified that the x-axis represents factor variables for individual years, aligned with our Generalized Additive Model (GAM), which treats 'fishing season' as a discrete variable for daily fishing data analysis. This differs from the continuous variables like Sea Surface Temperature (SST), tidal range, and lunar phase, which are typically depicted with smooth curves. To eliminate any ambiguity, we will revise Figure 11, Panel A, to annotate each year more clearly under the x-axis, highlighting the distinct representation of 'fishing season' as separate points or bars for each year.

  • Additions to the Discussion Section:

   - Following your suggestion, we have expanded the discussion to include the condition of parent eels and the impact of fishery activities.

Regarding the Quality of English Language:

We have engaged Dr. Yan Jizhou, a seasoned researcher with over a decade of experience in biology, for meticulous language polishing and revision of our manuscript.

We are immensely grateful for your valuable comments and suggestions, which have significantly enhanced the quality of our research. We eagerly anticipate your further feedback to continue refining our study.

Yours sincerely,

Hongyi Guo on behalf of all authors.

Reviewer 2 Report

Comments and Suggestions for Authors

Comments to the Author

Manuscript Number - biology-2815150

The manuscript entitled “Temporal Pattern and Environmental Influences on Anguilla Japonica Recruitment in the Yangtze Estuary” is a nice piece of work done by the authors. In this manuscript, the author's study on Japanese eel populations in the Yangtze River estuary found a bimodal seasonal arrival pattern with a triennial cycle. The abundance was strongly correlated with optimal temperatures and spring tides. The study's insights are crucial for developing conservation measures and preserving the ecological balance of the river. The findings are vital for understanding the decline of the species.

Overall, the design and execution of this work are excellent and nicely documented, the manuscript is well written, and the study's objectives are adequately described, and findings are effectively addressed. Thus, this manuscript is suitable for publishing.

Author Response

Dear Reviewer,

We are deeply thankful for your constructive and encouraging feedback on our manuscript "Temporal Pattern and Environmental Influences on Anguilla Japonica Recruitment in the Yangtze Estuary," identified by the manuscript number biology-2815150. It is particularly gratifying to note your recognition of our work as a meaningful contribution to conservation efforts and the understanding of species population trends.

Your appreciation for the clarity of our manuscript, the comprehensive approach to our research design and implementation, and the effective presentation of our objectives and results is highly valued. Knowing that our manuscript has met your approval for publication is indeed fulfilling.

We concur with your view that the insights from our study are vital for formulating conservation policies and for the preservation of the river's ecological integrity. We are dedicated to continuing our research in this domain, with the aspiration that our findings will significantly benefit the comprehension and management of Anguilla Japonica populations.

Once again, we extend our sincere thanks for your supportive comments. We are looking forward to the opportunity to contribute to the scholarly literature with our publication.

Sincerely,

Hongyi Guo on behalf of all the authors.

Reviewer 3 Report

Comments and Suggestions for Authors

The stock dynamics of Japanese eel in the Yangtze River estuary, where declining stocks are a problem, were clarified. This study has information that contributes to the resource management of this species and is deemed worthy of publication. Some minor corrections are required and I would like to see a revision by the authors.

Major comments

1: The latter part of the summary involves ambiguous content.

Write in the part of the abstract about the negative effects of rising water temperatures on eel recruitment, as shown by the ARIMA model of L446-458. This would increase the value of this paper.

2: The topic of El Niño is abruptly mentioned in L104-105.

Cite reference no. 37 here and explain why the focus is on El Niño.

Minor comments

Keywords should be corrected in alphabetical order.

Many places with no space between numbers and units.

Spaces before and after > and ± symbols should also be inserted.

Figure 1: Increase the size of the text (country name) in the top right-hand panel.

Figure 2 is clearly less complete in this paper.

Place the second string from the top in front of the yellow frame.

The division between 1 mm mesh and 3 mm mesh is a straight line.

Express it three-dimensionally.

Name the parts of the fishing gear.

Figure 3:

Note the number of samples.

 Border the columns with black.

 Add the year at which the samples were obtained in the figure legend.

Figure 4:

 Note the number of samples.

 Add the year at which the samples were obtained in the figure legend.

Figure 5:

 Note the number of samples.

 Border the columns with black.

 Add the year at which the samples were obtained in the figure legend.

Small capitals on the X axis should be in regular capital letters.

Figure 6:

 Match the range of values on the X and Y axes for all panels.

The color scheme of the full moon and the new moon should be reversed.

Water temperature graph should be attached above the each panels.

Figure 7:

 Border the columns with black.

Figure 8:

The yellow line on panel D is difficult to see. Make it a darker color.

Figure 11, Panel A:

Does the scale on the x-axis indicate January 1?

Put a scale at the January 1 position on the X-axis, and set a label indicating the year in between.

Figure 11, Panel C:

Full and new moon circles should be added.

Author Response

Dear Reviewer,

On behalf of my co-authors and myself, I would like to extend our heartfelt thanks for your thorough review of our manuscript titled "Temporal Patterns and Environmental Influences on Anguilla japonica Recruitment in the Yangtze Estuary." We are grateful for your recognition of the contribution our study offers to the resource management of this species and for endorsing our paper for publication.

We have meticulously considered each of your valuable suggestions and have undertaken a comprehensive revision of our manuscript. The revised sections have been highlighted in red for ease of identification. Below, we present our responses to your major and minor comments:

Major Comments:

1 Concerning the ambiguity in the latter part of the summary, you suggested we elaborate on the negative effects of rising water temperatures on eel recruitment as indicated by our ARIMA model in lines 446-458 to enhance the paper's value.

Response: We appreciate your insightful suggestion. We have amended the abstract to clearly state the negative impacts of increasing water temperatures on Japanese eel recruitment as demonstrated by our ARIMA model. The modifications are marked in red.

2 You mentioned that the discussion on El Niño is introduced abruptly in lines 104-105.

Response: We have now supplemented the introduction with a detailed explanation of El Niño's impact on recruitment, citing Reference No. 37, to clarify why El Niño is a focal point in our study.

Minor Comments:

1 Keywords should be listed in alphabetical order.

Response: Thank you for this reminder. The keywords have been reordered alphabetically as advised.

2 There are several instances where spaces are missing between numbers and units, as well as around ">" and "±" symbols.

Response: We are grateful for your attention to detail. Spaces have been added where necessary throughout the manuscript.

3 Figure 1: The text size (country name) in the top right-hand panel should be increased.

Response: As recommended, we have increased the text size in Figure 1.

4 Figure 2 appears incomplete.

Response: We have thoroughly revised Figure 2 to address completeness, including repositioning elements, clarifying divisions, adding three-dimensional effects, and labeling parts of the fishing gear as suggested.

5 5-7. For Figures 3, 4, and 5:

Response: We have annotated each figure with the number of samples, bordered the columns with black, and added the year of sample collection to the figure legends.

6 Figure 6:

Response: The value ranges on the X and Y axes have been matched for all panels. We have reversed the color scheme for the full moon and new moon symbols and attached a water temperature graph above each panel as suggested.

7 Figure 7:

Response: The columns have been bordered with black as recommended.

8 Figure 8:

Response: The yellow line on panel D has been changed to a darker color to enhance visibility, now appearing in coffee brown.

9 Figure 11, Panel A:

Response: We have clarified that the scale on the x-axis represents factor variables corresponding to individual years, not a continuous timeline starting from January 1st. The figure has been revised to more clearly denote each year under its respective category on the x-axis.

10 Figure 11, Panel C:

Response: Icons for the full moon and new moon have been added to Panel C of Figure 11 as suggested.

We thank you once again for your constructive feedback which has significantly improved our manuscript. We look forward to your final assessment.

Sincerely,
Hongyi Guo on behalf of all authors.